# Effects of Road Traffic on the Accuracy and Bias of Low-Cost Particulate Matter Sensor Measurements in Houston, Texas

**DOI:** 10.3390/ijerph19031086

**Published:** 2022-01-19

**Authors:** Temitope Oluwadairo, Lawrence Whitehead, Elaine Symanski, Cici Bauer, Arch Carson, Inkyu Han

**Affiliations:** 1Department of Epidemiology, Human Genetics, and Environmental Sciences, School of Public Health, University of Texas Health Science Center at Houston, Houston, TX 77030, USA; oluwadairot@gmail.com (T.O.); Lawrence.Whitehead@uth.tmc.edu (L.W.); Arch.Carson@uth.tmc.edu (A.C.); 2Center for Precision Environmental Health, Department of Medicine, Baylor College of Medicine, Houston, TX 77030, USA; elaine.symanski@bcm.edu; 3Department of Biostatistics, School of Public Health, University of Texas Health Science Center at Houston, Houston, TX 77030, USA; Cici.X.Bauer@uth.tmc.edu; 4Department of Epidemiology and Biostatistics, Temple University College of Public Health, Philadelphia, PA 19122, USA

**Keywords:** low-cost sensors, road traffic, particulate matter (PM), PM monitoring, PM sensor calibration

## Abstract

Although PM_2.5_ measurements of low-cost particulate matter sensors (LCPMS) generally show moderate and strong correlations with those from research-grade air monitors, the data quality of LCPMS has not been fully assessed in urban environments with different road traffic conditions. We examined the linear relationships between PM_2.5_ measurements taken by an LCPMS (Dylos DC1700) and two research grade monitors, a personal environmental monitor (PEM) and the GRIMM 11R, in three different urban environments, and compared the accuracy (slope) and bias of these environments. PM_2.5_ measurements were carried out at three locations in Houston, Texas (Clinton Drive largely with diesel trucks, US-59 mostly with gasoline vehicles, and a residential home with no major sources of traffic emissions nearby). The slopes of the regressions of the PEM on Dylos and Grimm measurements varied by location (e.g., PEM/Dylos slope at Clinton Drive = 0.98 (*R*^2^ = 0.77), at US-59 = 0.63 (*R*^2^ = 0.42), and at the residence = 0.29 (*R*^2^ = 0.31)). Although the regression slopes and coefficients differed across the three urban environments, the mean percent bias was not significantly different. Using the correct slope for LCPMS measurements is key for accurately estimating ambient PM_2.5_ mass in urban environments.

## 1. Introduction

Exposure to fine particulate matter (PM_2.5_) has been associated with several adverse health outcomes [1,2,3]. These health outcomes include increased hospitalizations, morbidity, and mortality from respiratory effects (infections, exacerbations of asthma, and chronic obstructive pulmonary disease) [4,5,6,7,8], and cardiovascular effects (ischemic heart disease, stroke) [4,5,6,8,9]. Due to the severity of these adverse health outcomes associated with PM_2.5_ exposures, monitoring PM_2.5_ for exposure assessment within communities has become a subject of interest among environmental scientists, governmental organizations, and interested private citizens. Recently, it has been suggested that the application of low-cost air sensors could improve the understanding of current indoor and ambient exposures to air pollutants, including PM_2.5_ [10,11,12,13]. Low-cost air sensors are portable and lightweight [11,13]. Furthermore, the direct reading capabilities of these sensors allow for instantaneous readings to obtain fine temporal resolution of the data that can be used for the evaluation of short-term exposure to air pollutants. For example, in the city of Houston (land area: 1651 km^2^), Texas (TX), only seven continuous air monitoring stations (CAMS) are currently operated for PM_2.5_ measurements by the Texas Commission on Environmental Quality (TCEQ) and U.S. Environmental Protection Agency (U.S. EPA). Low-cost particulate matter sensors (LCPMS) deployed in multiple locations can supplement the sparse ambient air PM_2.5_ data in cities such as Houston. Thus, finer scaled spatial data from LCPMS networks can result in more representative measurements and exposure assessments. The affordable LCPMS have also enabled individuals and private organizations to assess their own PM_2.5_ exposure with limited resources (e.g., budget or staff) [10,14].

Despite the advantage of using LCPMS for airborne PM_2.5_ monitoring, the data quality of measured LCPMS is open to question in comparison to the CAMS data. The LCPMS measurements must be validated before widespread use, in comparison with measurements from currently validated methods such as gravimetric methods. To this end, studies in the last decade have examined the validity of commercially available LCPMS [15,16,17,18,19,20,21,22]. While these studies found moderate to strong correlation (*r* range: 0.66–0.99) between the LCPMS and widely used research-grade monitors [12,15,19,23], the linear relationship (e.g., regression slopes) between the LCPMS and the research grade monitors can be affected in different environments [15,19].

The physical characteristics of ambient PM in urban environments are determined by various emission sources close to the location. For example, PM near major roadways contains submicron sizes of carbonaceous aerosols emitted from vehicle exhausts [24,25]. On the other hand, PM near steel plants is composed of metal aerosols with larger particle sizes from metal grinding and cutting processes [26]. Since most LCPMS count the number of particles in the air passing through the sensor, the reported concentrations of PM are dependent on the physical size and chemical characteristics of the aerosol [12,15,19]. Although the association between an LCPMS (Dylos air quality monitor) and a research-grade direct reading instrument (i.e., TSI Sidepak AM-510) has been assessed in indoor environments with various emission sources [27], few studies have investigated the association between LCPMS and research-grade direct reading instruments (DRI) under real-world ambient environments affected by varying traffic conditions. To be used as a supplementary PM measurement, it is important to determine whether the quality of the LCPMS data is reliable in urban environments with different traffic conditions. Therefore, the main objective of this study was to investigate the role of urban environments with different traffic conditions on the linear relationship between PM_2.5_ measurements taken by an LCPMS (Dylos DC1700) and gravimetric method as well as a research-grade DRI (GRIMM 11R). The field campaign was performed with a sampling duration of 3 h at three different locations in Houston, TX: (1) near a major road with large volume of heavy-duty diesel vehicles (HDDV); (2) near a major road with mostly gasoline vehicles; and (3) at a residential location with no major traffic related PM sources.

## 2. Materials and Methods

### 2.1. Equipment

The Dylos DC1700 air quality monitor (Dylos) (Dylos Corporation, Riverside, CA, USA) is a relatively inexpensive (<USD 500) and portable air monitoring sensor. The instrument uses a light scattering method to measure the number of particles in two size bins: >0.5 µm and >2.5 µm. The Dylos was selected due to the availability of research data from previous studies for comparison with this study.

The Grimm Mini Laser Aerosol Spectrometer 11R (Grimm) (GRIMM 11R) (Grimm Aerosol Technik GmbH & Co. KG, Ainring, Germany) was selected to compare with the Dylos measurements. The GRIMM 11R counts particles in the size range from 0.25 µm to 32 µm using a light scattering method. PM_2.5_ mass concentrations were obtained using Grimm Software 1178 V8-1 Rev IV. The cost of the Grimm 11R (USD 20,000) was about 40 times greater than the Dylos DC1700.

The Personal Environmental Monitor (PEM) Impactor (SKC Inc., Eighty-Four, PA, USA) was also used to compare with the Dylos data. The 37 mm polytetrafluoroethylene (PTFE) filter was placed in the PEM to collect PM_2.5_. The SKC Leland Legacy pump (SKC Inc., Eighty-Four, PA, USA) was used to pull air at 10 L/min. Unlike the Dylos and Grimm, PEM is a gravimetric method to measure time-integrated PM_2.5_ mass. In this study, the sampling time was 3 h at each location. The total cost of the PEM and the pump was about USD 1900.

The Thermo Orion Cahn-35 Microbalance (Thermo Electron Corporation, Beverly, MA, USA) was used to weigh the filters from the PEM. The accuracy of the gravimetric analysis as 0.0012% with a weight range and sensitivity of 250 mg/1.0.

The HOBO Data Logger U12-012 (Onset Computer Corporation, Bourne, MA, USA) was used to monitor ambient air temperature. The HOBO measures temperatures are in the range from −20 to 70 °C. The accuracy of HOBO temperature measurements is documented by the manufacturers as ±0.35 °C from 0 to 50 °C.

### 2.2. Sampling Locations

To determine the effect of traffic volumes on the linear relationship between PM_2.5_ measurements taken by the Dylos DC1700 (Dylos) and other research-grade instruments (PEM and GRIMM 11R), we simultaneously deployed a Dylos, a PEM, a Grimm 11R, and a HOBO at each sampling location (Figure 1a).
Clinton Drive, Houston, Texas (major road with a higher percentage of traffic emissions from heavy-duty diesel vehicles (HDDV)). Clinton Drive is located in the eastern part of the Houston metropolis in Texas (Figure 1b). In this study, the traffic on this road was made up of a higher number of HDDV (28%), emitting diesel particles. All samplers were deployed near the fence line of the Clinton CAMS, about twenty meters from Clinton Drive.US-59 Highway, Houston, Texas (major highway with traffic emissions mainly from gasoline vehicles (GV)). The US-59 highway in Houston, Texas runs from southwest to northeast in Houston (Figure 1c). The proportion of HDDVs among total traffic counts was 3% in this study. All samplers were deployed about 50 m from the road on a side street (Eastside Street).Residential home (residential location with no major sources of PM), which was located in a suburban area of Houston, Texas (Figure 1d). There were no significant sources of PM near the sample location. The closest major roadway from the residence was about 6400 m away, and there were no factories or industrial facilities close to the residence. All samplers were deployed in the backyard of the residence.

### 2.3. Study Design

One Dylos DC1700 (Dylos) was collocated with one PM_2.5_ PEM, one GRIMM 11-R (Grimm), and a HOBO data logger (HOBO) at all three locations. The Dylos, PEM, and Grimm measured ambient PM_2.5_ for 3 h over 20 days at each location. Sampling was conducted at one location each day, alternating between the three sampling locations. Samples were collected from October 2019 through January 2020. The HOBO was also deployed to record temperature during the 3 h sampling periods. The HOBO, Dylos, and Grimm were programmed to record data at 1 min intervals. Samples were collected in the mornings (8 a.m.–1 p.m.) on weekdays (Monday to Friday) and weekends (Saturday and Sunday). Samples were not collected when weather conditions were not conducive for sampling (e.g., on rainy days). A total of 53 valid sampling campaigns, 40 sample days on weekdays, and 13 sample days on weekends were obtained.

To estimate the ratio of HDDV to gasoline vehicles on the two major roads (US-59 and Clinton Drive), the number of each type of vehicle passing was counted using video recording during the sampling periods. Vehicles passing by were counted for a 10 min period in every hour. The total number of vehicles counted per hour was then multiplied by 6.

Prior to this study, the Dylos, Grimm, and HOBO were previously calibrated by the manufacturer. The PTFE filters were pre-weighed using the Cahn-35 microbalance (Thermo Electron Corporation, Beverly, MA, USA) in a weighing room at the University of Texas Health Science Center at the Houston (UTHealth) School of Public Health. The batteries for the SKC Leland pumps (SKC Inc., Eighty-Four, PA, USA), the Dylos DC1700, and the Grimm 11R were fully charged before the deployment. On each day of sampling, the Leland legacy pump was calibrated at 10 L/min using the BIOS Dry-Cal DC-lite (BIOS International Corp., Butler, NJ, USA) at the sampling locations. The clocks on the Dylos, Grimm, and HOBO were all synchronized with the internet clock on a laptop. The information on the sampling date, time, location, sample ID, pump ID, filters’ pre-weight, and pump pre-flow rates was recorded. The instruments were placed side by side on a portable folding table, which was about 1 m above the ground level (Figure 2). All the sampling instruments were turned on simultaneously, and the start time was recorded. After three hours, all the instruments were turned off, and the stop time was recorded. After sampling, the post-flow rate for PEM was measured using the Dry Cal DC-Lite. All the collected filters were transferred to the UTHealth School of Public Health for gravimetric analysis. The post-weight and the post-flow rate were recorded. The data logged on the Dylos, the Grimm 11R, and the HOBO data logger were downloaded onto a laptop computer.

### 2.4. Data Analysis

To obtain PM_2.5_ particle number concentrations from the Dylos, the number counts for particles > 2.5 µm were subtracted from the number counts for particles > 0.5 µm. The 3 h mean particle count concentrations for the Dylos and 3 h mean mass concentration for the Grimm for each sampling day were calculated using 1 min raw data from both instruments. To obtain the 3 h integrated PEM PM_2.5_ mass concentration for each sampling day, the total mass (post weight–pre weight) of PM_2.5_ on each filter was divided by the total volume of air (average flow rate × sampling duration, approximately 3 h) on the same sample day. The mean of the 3 h temperature was also calculated from the 1 min interval data from the HOBO. All the data were analyzed using STATA15 (StataCorp LLC., College Station, TX, USA).

A descriptive analysis was performed using 3 h mean PM_2.5_ concentrations and ambient temperature across three sampling locations. Simple linear regression analysis was conducted using overall data and location-specific data, separately. The effect of sampling location (as a proxy of different traffic emission sources) on the linear relationship between the Dylos DC1700 and PEM measurements was assessed with the linear regression model from Equation (1) below. Before the regression analysis was carried out, a natural log transformation of the data was carried out to fit the data close to a Gaussian distribution.
Y = β_0_ + β_1_ X_1_ + β_2_ X_2_ + β_3_ X_3_ + β_4_ (X_1_ × X_2_) + β_5_ (X_1_ × X_3_)(1)
where

Y = natural log of the 3 h PM_2.5_ mass concentration measured by the PEM or Grimm;X_1_ = natural log of the 3 h PM_0.5–2.5_ particle number concentration measured by the Dylos;X_2_ = binary dummy variable coded as 1 for US-59 and zero (0) for the other two locations (Clinton Drive and the residence); andX_3_ = binary dummy variable coded as 1 for the residence and zero (0) for all other locations (Clinton Drive and US-59).

In Equation (1), the Clinton Drive location, coded as zero for both X_2_ and X_3_, is the reference sampling location to which all the other locations (US-59 and Residence) were compared.

Two interaction terms (X_1_ × X_2_ and X_1_ × X_3_) were introduced into a linear regression to test for the difference in slopes across sampling locations. The null hypothesis for the model that the difference between the slope at Clinton Drive and the slope at US-59 (represented by the β_4_ coefficient) equal to zero was tested. Similarly, a null hypothesis was tested whereby the difference between the slope at Clinton Drive and the slope at the residence (represented by the β_5_ coefficient) was equal to zero.

To test for the difference between US-59 and the residence, the coding for the variables in the regression analysis from Equation (1) was slightly modified. The residence location was recoded as zero for both X_2_ and X_3_ (the reference sampling location). The X_3_ was recoded as (1) for Clinton and (0) for all other locations, while the X_2_ variable remained coded as (1) for US-59 and (0) for all other locations. The X_1_ variable remained the same. The β_4_ coefficient for this modified model was tested for the difference between the slopes of US-59 and the residence.

Further analysis was carried out to determine the effect of other covariates such as temperature and the ratio of heavy-duty trucks to gasoline cars (HDDV%) on the linear relationship between the Dylos and the each of the research grade instruments.

The agreement (or bias) between the Dylos DC1700 measurements and the research-grade instruments across sampling locations was assessed by calculating the mean absolute relative error. To estimate the absolute relative error, the Dylos number concentration measurements were first converted to mass concentration measurements using fitted regression line equations from a linear regression of (i) the PEM on the Dylos measurements and (ii) the Grimm on the Dylos measurements. This conversion was carried out using two methods.

General equation (GE) method: A single fitted regression line equation from the linear regression of the total combined data was obtained and used to convert the Dylos PM_2.5_ measurements from all three locations.Sampling location equation (SLE) method: A different regression line was constructed, stratified by each sampling location. Three fitted regression equations, one for each sampling location (Clinton, US-59, and the residence), were used to convert the Dylos PM_2.5_ measurements.

After obtaining the converted Dylos mass concentrations, the absolute relative error was calculated for each day of sampling with Equation (2) below. The mean absolute relative error (MARE) was then estimated by calculating the average of all the absolute relative errors for all the sampling days. The data were then grouped by the sampling location, and the MARE for each location was calculated. With the equal variance structure, an analysis of variance (ANOVA) test with Bonferroni correction was used to compare the difference of the MAREs between the Dylos–Grimm at the three locations. For the Dylos–PEM comparison, because the equal variance between groups assumption was unmet, two-sample t-tests were used to compare the difference between the MAREs across the three locations.
|(Dylos PM_2.5_ − PEM (or Grimm) PM_2.5_)|/(PEM (or Grimm) PM_2.5_)(2)
where

Dylos PM_2.5_ = converted PM_2.5_ mass concentrations from the 3 h mean Dylos count measurements collected over a single sample duration; andPEM or Grimm PM_2.5_ = 3 h integrated PM_2.5_ mass concentration collected by the PEM or Grimm over a single sample duration.

## 3. Results

### 3.1. Statistical Summary

A total of 53 valid sampling days was obtained from the Dylos DC1700 after excluding seven outliers in this study. Outliers were identified as data points with residuals greater than two standard deviations. The mean and standard deviation (SD) of the particle count for the Dylos was 1439 ± 1053 particles/0.01 ft^3^ (Range: 158–4394 particles/0.01 ft^3^). The means and SDs of the PEM and the Grimm 11-R were 24.4 ± 24.4 µg/m^3^ (Range: 5.1–137.8 µg/m^3^) and 13.7 ± 10.7 µg/m^3^ (Range: 1.9–47.6 µg/m^3^), respectively. The mean ambient air temperature during the entire study period was 25.0 ± 6.5 °C (range: 10.9–37.3 °C).

The descriptive statistics by sampling locations are summarized in Table 1. The mean PM_2.5_ count concentration from the Dylos did not differ by sampling location (ANOVA *p*-value = 0.33). The mean particle count at the Clinton Drive, US-59, and residence locations were 1737 ± 1178 particles/0.01 ft^3^, 1235 ± 854 particles/0.01 ft^3^, and 1332 ± 1082 particles/0.01 ft^3^, respectively. The mean PM_2.5_ mass concentration from the PEM (ANOVA *p*-value = 0.01) and the Grimm (ANOVA *p*-value = 0.03) differed by sampling location. The mean mass concentration from the PEM was 39.93 ± 36.81 µg/m^3^ at Clinton Drive, 18.9 ± 9.9 µg/m^3^ at US-59, and 15.2 ± 5.6 µg/m^3^ at the residence. The 3 h mean mass concentration from the Grimm was 19.0 ± 14.7 µg/m^3^ at Clinton Drive, 10.4 ± 5.2 µg/m^3^ at US-59, and 11.6 ± 7.8 µg/m^3^ at the residence. The 3 h mean temperature was 27.3 ± 5.2 °C, 21.3 ± 5.9 °C, and 26.3 ± 7.1 °C at the Clinton, US-59, and residence locations, respectively.

### 3.2. Effect of Road Traffic as a Proxy of PM_2.5_ Emission Source on the Linear Relationship between Dylos and Research Grade Instruments (PEM and Grimm 11R)

Dylos vs. PEM. The slope for the overall combined log-transformed data was 0.70 (*R*^2^ = 0.52). The slope comparing the Dylos to the PEM measurements was 0.98 (*R*^2^ = 0.77), 0.63 (*R*^2^ = 0.42), and 0.29 (*R*^2^ = 0.31) at the Clinton Drive, US-59, and the residence, respectively (Figure 3).

We assessed the significance of the coefficient of the interaction terms β_4_ (Clinton Drive vs. US-59) and β_5_ (Clinton Drive vs. Residence) from Equation (1). While the β_4_ coefficient (−0.36) was not significantly different (*p* = 0.89) between Clinton and US-59, the β_5_ coefficient (−0.69) was statistically different between Clinton and the residence (*p* < 0.01). There was no significant difference between the slopes of US-59 and the residence (Coefficient = 0.33, *p* = 0.13).

Dylos vs. Grimm. Figure 4 shows the results of the linear regression of the log-transformed Grimm measurements on the log-transformed Dylos measurements. The slope for the overall combined data was 0.91 (*R*^2^ = 0.86). The slopes comparing the Dylos to the Grimm measurements were 1.10 (*R*^2^ = 0.95), 0.73 (*R*^2^ = 0.79), and 0.76 (*R*^2^ = 0.83) at Clinton Drive, US-59, and the residence, respectively.

Similarly, we assessed the significance of the coefficient of interaction terms β_4_ (Clinton vs. US-59) and β_5_ (Clinton vs. residence) for the Dylos–Grimm regression, β_4_ (coefficient = −0.37, *p* < 0.01), and β_5_ (coefficient = −0.33, *p* < 0.01). The coefficient of the interaction term (−0.03) comparing US-59 to the residence was not statistically different (*p* = 0.80).

### 3.3. Effect of Temperature and Truck Ratio (HDDV%) on the Linear Relationship between Dylos and Research Grade Instruments (PEM and Grimm 11R)

For the PEM, the coefficient for temperature (0.023) in the final model was significant, indicating an increase of 1.26 µg/m^3^ in PEM mass concentration measurements with a 10 °C rise in temperature. The effect of truck ratio was not significant on the relationship between Dylos and PEM. A similar finding was obtained with the Grimm and Dylos. The coefficient for temperature (0.030) indicated an increase of 1.35 µg/m^3^ in Grimm mass concentration measurements with a 10 °C rise in temperature. The slopes at three sampling locations and the difference in slopes by location between without temperature (Model 1) and with temperature (Model 2) are summarized in Table 2. For PEM, ambient temperature increased regression slope for US-59 (0.63 to 0.82), whereas it did not change the slopes for other locations. Similarly, we found that the regression slope for Grimm was incresed from 0.73 (Model 1) to 0.84 (Model 2) for US-59, while they were not significantly changed between Model 1 and Model 2 for other locations.

### 3.4. Agreement between the Dylos and the Reference Grade Instruments (PEM and Grimm 11R)

The bias between the Dylos and the PEM PM_2.5_ measurements was about two times greater than the bias between the Dylos and the Grimm (Table 3). Using the GE method, for instance, the MARE between the Dylos and PEM measurements for the combined data was 42 ± 35%, whereas the MARE between the Dylos and Grimm measurements was 22 ± 16%.

Table 3 also summarizes the MARE using the SLE method. For Dylos to PEM PM_2.5_ measurements, the MAREs were greater at Clinton (37%) and US-59 (37%) than at the residence (27%). However, the MAREs across sampling locations were not statistically different from each other (*p*-values = 0.89 (Clinton vs. US-59), 0.22 (Clinton vs. the residence), and 0.39 (US-59 vs. the residence). The MAREs using the SLE method between the Dylos and Grimm at all locations were similar (Clinton: 14%, US-59: 19%, and the residence: 19%), and the *p* values for all comparisons were 1.0.

Figure 5 shows temporal trends between the instruments on weekdays (Monday through Friday) and weekends (Saturday and Sunday). The 3 h mean PM_2.5_ concentrations from all three instruments show similar temporal trends across different sampling days. The peak 3 h mean PM_2.5_ concentrations were much larger on weekdays compared to weekends. The MARE (estimated using the SLE) between the Dylos and PEM PM_2.5_ was similar on weekdays (32 ± 32%) and weekends (36 ± 30%). The MARE between the Dylos and the Grimm was also similar on weekdays (18 ± 14%) and weekends (15 ± 12%).

## 4. Discussion

The effects of different types of road traffic on the linear relationship between the Dylos DC1700 and the two research-grade instruments (PEM and Grimm 11R) were assessed at three specific locations with varying traffic types. The agreement (or bias) between the Dylos and the research-grade instruments was also examined. Environments with different traffic characteristics significantly changed the regression slopes between the Dylos DC1700 and the other instruments. However, the agreement (or bias) between the instruments was not significantly affected by traffic conditions.

The 3 h integrated PM_2.5_ mass concentration measured by the PEM at the Clinton Drive location (HDDV% = 28 ± 11%) was at least two times higher than those at the US-59 highway (HDDV% = 3.3 ± 0.7%) and the residence (no HDDV). Although both the Clinton Drive and the US-59 locations were in close proximity to busy roadways, one possible explanation for the higher concentration of PM_2.5_ at Clinton Drive than the US-59 may be associated with more resuspension of PM_2.5_ at Clinton Drive. Askariyeh et al. (2020) observed that PM_2.5_ previously deposited on roadways was heavily resuspended due to the movement of heavy-duty vehicles (i.e., trucks) over the road. Estimated traffic-related PM_2.5_ increased up to 208% on roadways, as resuspended particles were included in the model [28]. A larger proportion of heavy vehicles on a road is also a factor increasing the rate of resuspended particles [28]. Hence, the elevated PM_2.5_ concentration at the Clinton Drive may be explained by the higher percentage of heavy-duty trucks in this study.

The regression of the log-transformed PEM over the log-transformed Dylos PM_2.5_ concentrations for the overall combined data showed positive and moderate to strong correlation (r = 0.72). Previous studies showed a similar correlation between a DRI and gravimetric samplers (Kim et al., 2004: r = 0.68 and Zhu et al., 2011: r = 0.63) [29,30]. We also found that the correlation (r = 0.95) between the Dylos and the Grimm 11-R was stronger than that between the Dylos and the PEM. The higher correlation between the Dylos and Grimm is probably due to the similar light scattering method used by both samplers as opposed to the gravimetric method used by the PEM. Previous studies observed a strong correlation between an LCPMS and other direct reading instruments (Han et al., 2017: r = 0.88 and Northcross et al., 2013: r = (0.90–0.99) [19,23].

Slopes from the linear regression equations obtained from collocated measurements of the Dylos and research-grade instruments are often used to estimate correction factors in the conversion of Dylos particle count concentration (particles count/0.01 ft^3^) to mass concentration (µg/m^3^). Converting the Dylos counts to mass concentration enables equivalent comparison between the Dylos measurements, other research-grade instruments, and the national air quality standards. The slope of simple linear regression of the log transformed PEM and Dylos data for the overall data was 0.70, whereas the slope of the log transformed Grimm11R and Dylos was 0.91. Using the linear regression equations in this study, the estimated Dylos PM_2.5_ for a particle count of 10,000 particles/0.01 ft^3^, for instance, was 82.9 µg/m^3^ using the PEM model and 76.8 µg/m^3^ using the Grimm 11R model. The results of this study overestimated PM_2.5_ approximately 20–30 percent compared to the PM_2.5_ for a 10,000 particle/0.01 ft^3^ in previous studies: 62.1 µg/m^3^ [22] or 67.8 µg/m^3^ [23].

Additionally, the slope of the Dylos–PEM varied by location. The discrepancies in regression slopes may be explained by the heterogeneous characteristics of aerosols at the different sampling locations. The PM_2.5_ at Clinton Drive was composed of a relatively higher percent of diesel particle emissions (HDDV = 28%) compared to US-59 (HDDV = 3%) and the residential location (no HDDV). The PM_2.5_ at the residence consisted of particles from suburban background sources with no significant traffic sources nearby. Given the different slopes by locations, sampling location specific conversion factors (SLE slopes) can be helpful in improving the accuracy of the estimated PM_2.5_ mass concentration from the Dylos PM_2.5_ count measurements (and other LCPMS) at locations where the characteristics of PM_2.5_ differ.

The effects of temperature and truck ratio on the linear relationship between the Dylos and the other research grade instruments were examined using the regression model (Model 2). The effect of truck ratio (ratio of heavy-duty trucks to regular gas vehicles) was not significant. Although temperature was a significant factor in both the Dylos–PEM and Dylos–Grimm linear relationship, the effect size of temperature was minimal compared to the location variable. The inclusion of temperature in the regression model modified less than 10 percent of the overall slope and the slopes at each specific sampling location. Only one exception was the difference between the US-59 and residence slopes which became statistically significant for the Dylos–PEM relationship.

The mean percentage bias as MARE between the Dylos and the PEM (42%) with the combined data in this study was similar to a previous study (40%) [29]. Further analysis of the MARE by sampling location showed that the mean percent bias did not significantly differ by sampling location. The results suggest that, in urban environments similar to Houston, TX, the bias (or precision) was consistent between the converted Dylos PM_2.5_ mass and the PM_2.5_ measured from the research-grade instruments.

The limitations of this study include the ambient PM_2.5_ at the sampling locations not being fully characterized for their composition aerosols. Chemical characterization of the PM_2.5_ at the sampling locations would help show how chemical constituents in PM_2.5_ affect the relationship between the LCPMS and research-grade instruments. Further, a longer sampling period (i.e., >30 days) at each location would be preferable to the 18 days of sampling in this study, as this would have provided a larger sample size to make stronger statistical conclusions. This knowledge would enable better conclusions about the results of the study.

## 5. Conclusions

The Dylos had a moderate to strong correlation with both the PEM and the Grimm. The slopes of the linear regression comparing the Dylos to both the PEM and the Grimm varied by about 30 percent across sampling locations. This finding suggests that calibration at each specific sampling location with different traffic characteristics is key to determine the PM_2.5_ mass concentrations using an LCPMS. On average, the bias between the converted Dylos measurements and the PEM measurements was about two times higher than the bias between the Dylos and the Grimm measurements. The biases between the Dylos and both research-grade instruments remained similar across different sampling locations and PM concentration levels. Overall, the LCPMS, after proper calibration, could provide supplementary PM_2.5_ data to evaluate air pollution in urban environments with different traffic characteristics.

## Figures and Tables

**Figure 1 ijerph-19-01086-f001:**
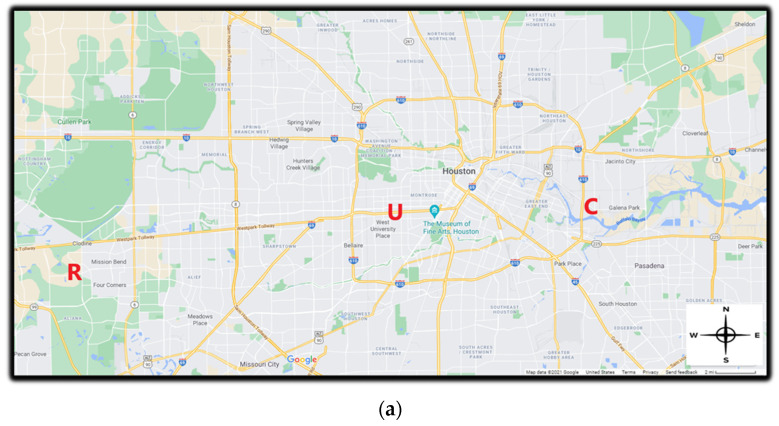
Map showing sampling locations. (**a**) All 3 sampling locations; (**b**) Clinton drive sampling location (Latitude: 29.7340345, Longitude: −95.2581138); (**c**) US—59 sampling location (Latitude: 29.731556, Longitude: −95.424426); (**d**) Residence sampling location (Latitude: 29.6854160, Longitude: −95.6937728). C = Clinton Drive; U = US-59; R = Residence. Map source: Google Maps.

**Figure 2 ijerph-19-01086-f002:**
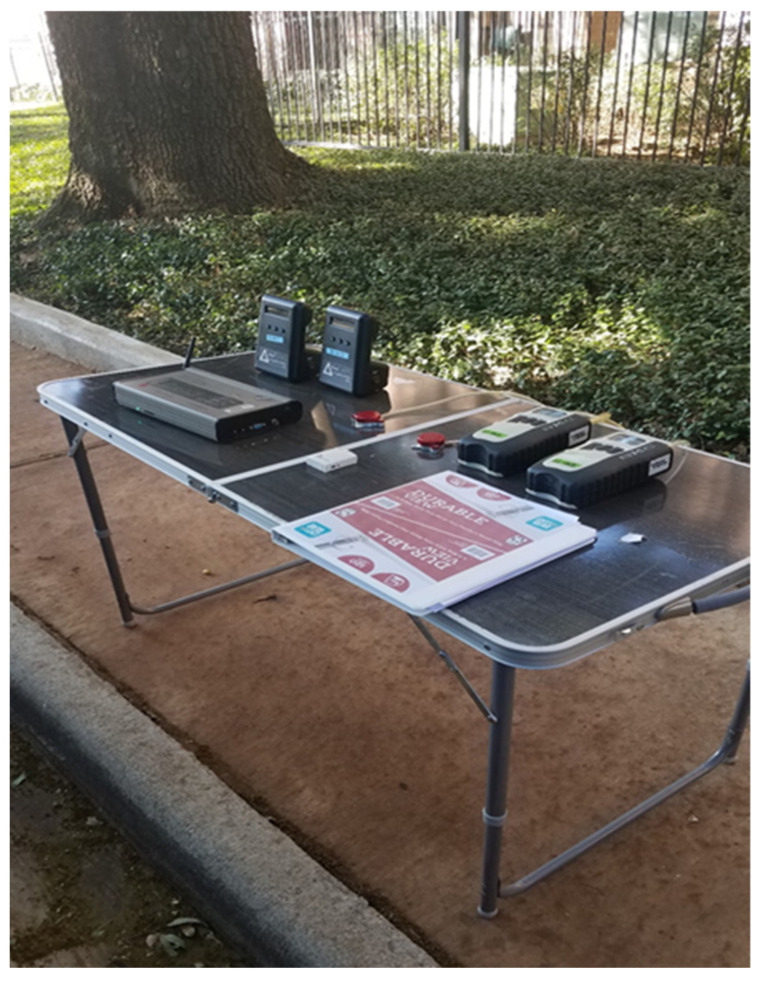
Sampling setup at US-59 sampling location.

**Figure 3 ijerph-19-01086-f003:**
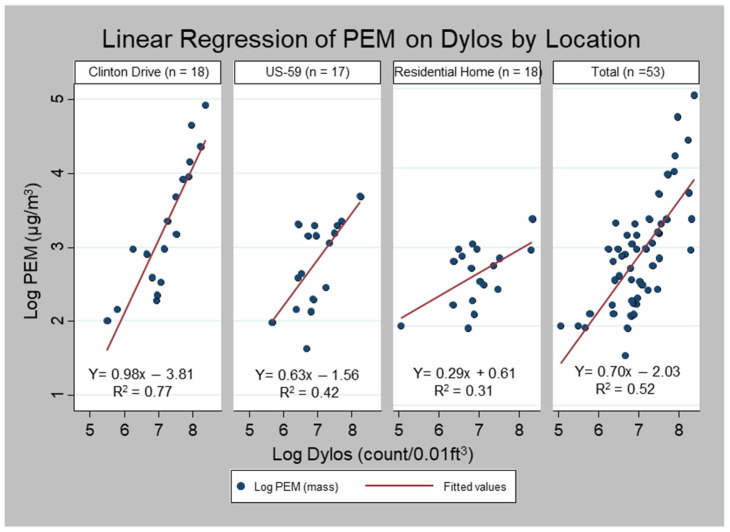
Linear regression of Dylos on PEM PM_2.5_ measurements by sampling location.

**Figure 4 ijerph-19-01086-f004:**
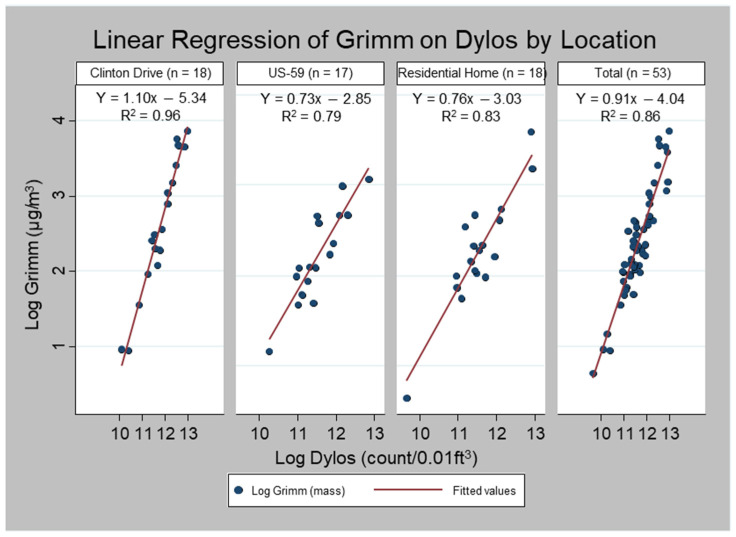
Linear regression of Dylos on Grimm PM_2.5_ measurements by sampling location.

**Figure 5 ijerph-19-01086-f005:**
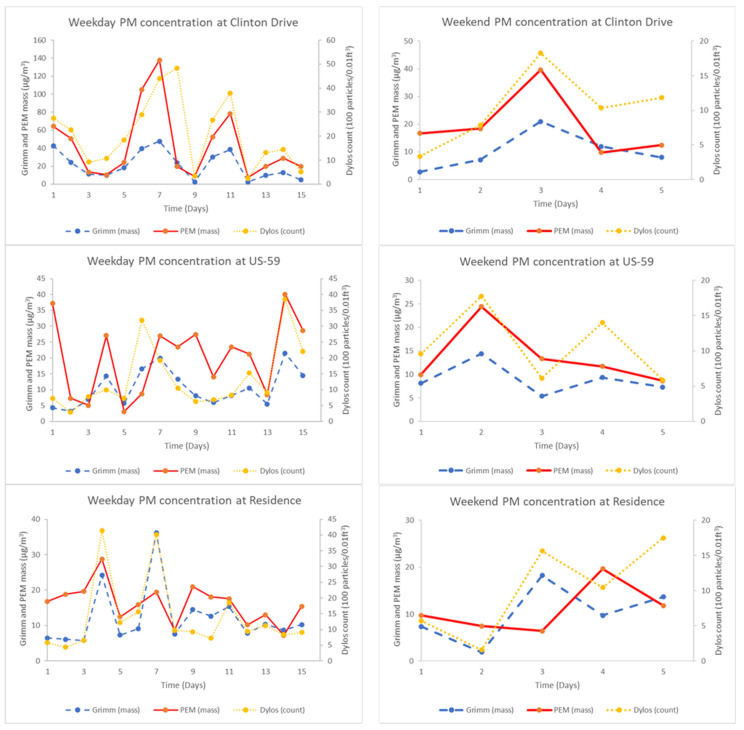
Comparison of 3 h mean PM_2.5_ concentration by weekdays and weekends. Weekdays include all days from Monday through Friday while weekend includes Saturday and Sunday only. Dylos PM_2.5_ counts units is 100 particles/0.01 ft^3^.

**Table 1 ijerph-19-01086-t001:** Summary of data obtained from the Dylos, Grimm, and HOBO by location.

Location	Instrument	Measurement	Sampling Days	Mean ± SD ^a^	Median	Min, 25% ^b^, 75% ^c^, Max
Clinton Drive	PEM	PM mass (µg/m^3^)	18	39.9 ± 36.8	21.9	7.4, 12.5, 52.5, 137.8
	Grimm 11R	PM mass (µg/m^3^)	18	19.0 ± 14.7	12.5	2.6, 8.0, 30.2, 47.6
	Dylos 1700	PM number (particles/0.01 ft^3^)	18	1737 ± 1178	137.6	246, 920, 2680, 4394
	HOBO	Temp (°C)	18	27.3 ± 5.2	28.0	13.4, 24.3, 30.6, 37.0
US-59	PEM	PM mass (µg/m^3^)	17	18.9 ± 9.9	21.3	5.1, 10.0, 27.0, 40.1
	Grimm 11R	PM mass (µg/m^3^)	17	10.4 ± 5.2	8.2	3.2, 6.9, 14.3, 21.5
	Dylos 1700	PM number (particles/0.01 ft^3^)	17	1235 ± 854	95.7	289, 957, 1529, 3844
	HOBO	Temp (°C)	17	21.3 ± 5.9	22.1	10.9,17.3, 25.5, 32.6
Residence	PEM	PM mass (µg/m^3^)	18	15.2 ± 5.6	15.7	7.2, 10.2, 19.5, 28.8
	Grimm 11R	PM mass (µg/m^3^)	18	11.6 ± 7.8	9.4	1.9, 7.4, 13.7, 36.2
	Dylos 1700	PM number (particles/0.01 ft^3^)	18	1332 ± 1082	95.4	158, 723, 1560, 4144
	HOBO	Temp (°C)	17 *	26.3 ± 7.1	26.6	12.7, 26.6, 30.9, 37.3

^a^ SD = standard deviation, ^b^ 25% = 25th percentile, ^c^ 75% = 75th percentile; * temperature data for 1 out of the 18 sampling days was missing due to technical issues with HOBO.

**Table 2 ijerph-19-01086-t002:** Comparison of slopes obtained from regression models.

**PEM**		**Model 1 ^a^**	**Model 2 ^b^**
Slope	Total	0.70	0.68
	Clinton	0.98	0.93
	US-59	0.63	0.82
	Residence	0.29	0.28
Slope difference	Clinton vs. US-59	−0.35 (*p* = 0.10)	−0.12 (*p* = 0.54)
	Clinton vs. Residence	−0.69 (*p* < 0.01)	−0.59 (*p* < 0.01)
	US-59 vs. Residence	0.33 (*p* = 0.13)	0.47 (*p* = 0.03)
*R* ^2^		0.68	0.74
**GRIMM**		**Model 1**	**Model 2**
Slope	Total	0.91	0.89
	Clinton	1.10	1.03
	US-59	0.73	0.84
	Residence	0.76	0.77
Slope difference	Clinton vs. US-59	−0.37 (*p* = 0.03)	−0.20 (*p* = 0.05)
	Clinton vs. Residence	−0.34 (*p* = 0.02)	−0.25 (*p* = 0.01)
	US-59 vs. Residence	−0.03 (*p* = 0.80)	−0.05 (*p* = 0.62)
*R* ^2^		0.90	0.94

^a^ Model 1: model including log-transformed dylos count and sampling location as varaibles, ^b^ Model 2: model including log-transofrmed dylos count, sampling location, and ambient temperature as varaibles.

**Table 3 ijerph-19-01086-t003:** Mean absolute relative error between Dylos and research grade instruments by location.

Location	Dylos vs. PEM (Mean (%) ± SD)		Dylos vs. Grimm (Mean (%) ± SD)		PEM vs. Grimm (Mean (%) ± SD)	
	^a^ GE	^b^ SLE	^a^ GE	^b^ SLE	^a^ GE	^b^ SLE
Clinton (*n* = 18)	38 ± 22	37 ± 33	19 ± 13	14 ± 13	36 ± 23	35 ± 36
US-59 (*n* = 17)	38 ± 45	37 ± 43	24 ± 17	19 ± 13	32 ±35	31 ± 33
Residence (*n* = 18)	51 ± 35	27 ± 21	22 ± 19	19 ± 16	42 ± 39	25 ± 21
^c^ Combined (*n* = 53)	42 ± 35	34 ± 33	22 ± 16	17 ± 14	37 ± 33	30 ± 30

^a^ Absolute relative error estimated from a single regression line equation of total combined data. GE = general equation; ^b^ absolute relative error estimated from 3 regression line equations of data after grouping by sampling location. SLE = sampling location equation; ^c^ absolute error for all sampling locations combined.

## Data Availability

The data presented in this study are available on request from the corresponding author.

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
