# Peer review of "Effects of Road Traffic on the Accuracy and Bias of Low-Cost Particulate Matter Sensor Measurements in Houston, Texas"

_ijerph, 2022, doi:10.3390/ijerph19031086_

Round 1
Reviewer 1 Report
Dear Authors,
thank you for the manuscript 'Effects of Road Traffic on the Accuracy and Bias of Low-Cost 2 Particulate Matter Sensor Measurements in Houston, Texas'.
The submitted document is well structured and can be read fluently. Results and conclusions are (in most parts) clear to me as a reviewer. In order to remove unclear issues I'd ask for a few changes and additions to be addressed in the manuscript:
- What is the accuracy of LOBO tempeature logger? The given temperature range is -20 to 700°C - a maximum value that I doubt (typo?).
- Did you record windspeed and esp. humidity?
- How were the sensor inlets oriented (vertically/horizontally)?
- Was there an influence of sunlight during the 3h-period of a measurement? In Fig. 2 it seems, the equiment was standing in the shadow of a tree. How about other locations?
- You are using mainly metric values in your manuscript. The only exception is the particle concentration of the Dylos sensor. Can you recalculate these values, please?
- Please give geographic coordinates of sampling locations
- Did you count the ratio or number of diesel and gasoline vehicles ?
- Did you measure baselines for anthropogenic aerosols?
- What is the given accuracy of the gravimetric analysis?
Thank you.
Reviewer 2 Report
The study between low-cost sensors and research-grade equipment is a very important topic for local traffic emission studies. These low-cost sensors can be placed in many traffic locations due to the lower cost in comparison to more expensive equipment. However, the sampling days could be longer than 18 days, maybe a full month of data would be more sufficient. Overall, the length and quality of this paper are good, and the research method is conducted correctly.
The author should consider increasing the period of data collection from 18 days to 30 days or above. The content of the article can be improved significantly with more available data to confirm the accuracy of the low-cost sensors.
Reviewer 3 Report
This manuscript just compared the examined the linear relationships between PM2.5 measurements taken by an LCPMS (Dylos DC1700) and two research grade monitors, a Personal Environmental Monitor (PEM) and 19 the GRIMM 11R, in three different urban environments, and compared the accuracy (slope) and bias in Houston, Texas. The study may have oversimplified the working conditions, various factors should be considered like temperature, velocity, etc.
Round 2
Reviewer 3 Report
the quality of manuscript has been greatly improved, and satisfy the stand of the journal. therefore, the manuscript can be accepted.